# Learning to Generate Questions by Recovering Answer-containing Sentences

## Abstract

To train a question answering model based on machine reading comprehension (MRC), significant effort is required to prepare annotated training data composed of questions and their answers from contexts. To mitigate this issue, recent research has focused on synthetically generating a question from a given context and an annotated (or generated) answer by training an additional generative model, which can be utilized to augment the training data. In light of this research direction, we propose a novel pre-training approach that learns to generate contextually rich questions, by recovering answer-containing sentences. Our approach is composed of two novel components, (1) dynamically determining $K$ answers from a given document and (2) pre-training the question generator on the task of generating the answer-containing sentence. We evaluate our method against existing ones in terms of the quality of generated questions as well as the fine-tuned MRC model accuracy after training on the data synthetically generated by our method. Experimental results demonstrate that our approach consistently improves the question generation capability of existing models such as T5 and UniLM, and shows state-of-the-art results on MS MARCO and NewsQA, and comparable results to the state-of-the-art on SQuAD. Additionally, we demonstrate that the data synthetically generated by our approach is beneficial for boosting up the downstream MRC accuracy across a wide range of datasets, such as SQuAD-v1.1, v2.0, and KorQuAD, without any modification to the existing MRC models. Furthermore, our experiments highlight that our method shines especially when a limited amount of training data is given, in terms of both pre-training and downstream MRC data.

## 1 Introduction

Machine reading comprehension (MRC), which finds the answer to a given question from its accompanying paragraphs (called context), is an essential task in natural language processing. With the release of high-quality human-annotated datasets for this task, such as SQuAD-v1.1 (Rajpurkar et al., 2016), SQuAD-v2.0 (Rajpurkar et al., 2018), and KorQuAD (Lim et al., 2019), researchers have proposed MRC models even surpassing human performance. These datasets commonly involve finding a snippet within a context as an answer to a given question.

However, these datasets require significant amount of human effort to create questions and their relevant answers from given contexts. Often the size of the annotated data is relatively small compared to that of data used in other self-supervised tasks such as language modeling, limiting the accuracy.

To overcome this issue, researchers have studied models for generating synthetic questions from a given context along with annotated (or generated) answers on large corpora such as Wikipedia. Golub et al. (2017) suggest a two-stage network of generating question-answer pairs which first chooses answers conditioned on the paragraph and then generates a question conditioned on the chosen answer. Dong et al. (2019) showed that pre-training on unified language modeling from large corpora including Wikipedia improves the question generation capability. Similarly, Alberti et al. (2019) introduced a self-supervised pre-training technique for question generation via the next-sentence generation task.

However, self-supervised pre-training techniques such as language modeling or next sentence generation are not specifically conditioned on the candidate answer and instead treat it like any other

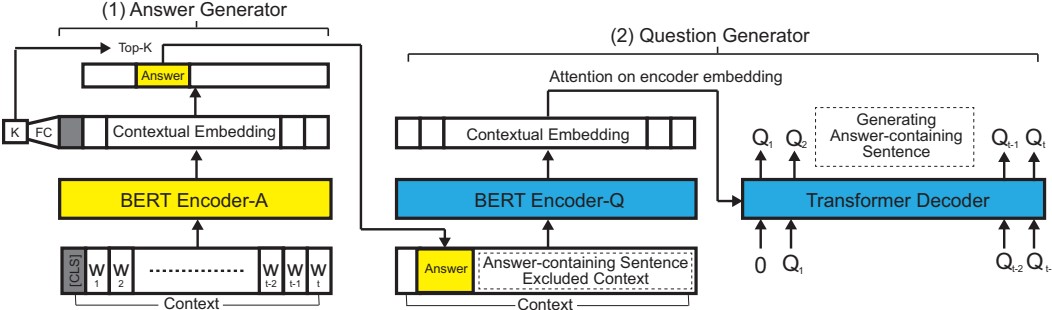

Figure 1: Architecture of a simple generative model, BertGen. When applying our training method "ASGen" to the model, the question generator takes as input the answer and the context with the answer-containing sentence removed and generates the missing answer-containing sentence.

phrase, despite the candidate answer being a strong conditional restriction for the question generation task. Also, not all sentences from a paragraph may be relevant to the questions or answers, so task of their generation may not be an ideal candidate as a pre-training method for question generation tasks. Moreover, in question generation it is important to determine which part of a given context can be a suitable answer for generating questions.

To address these issues, we propose a novel training method called Answer-containing Sentence Generation (ASGen) for a question generator. ASGen is composed of two steps: (1) dynamically predicting $K$ answers to generate diverse questions and (2) pre-training the question generator on the answer-containing sentence generation task. We evaluate our method against existing ones in terms of the generated question quality as well as the fine-tuned MRC model accuracy after training on the data synthetically generated by our method.

Experimental results demonstrate that our approach consistently improves the question generation quality of existing models such as T5 (Raffel et al., 2020) and UniLM (Dong et al., 2019), and shows state-of-the-art results on MS MARCO (Nguyen et al., 2016), NewsQA (Trischler et al., 2017), as well as comparable results to the state-of-the-art on SQuAD. Additionally, we demonstrate that the synthetically generated data by our approach can boost up downstream MRC accuracy across a wide range of datasets, such as SQuAD-v1.1, v2.0, and KorQuAD, without any modification to the existing MRC models. Furthermore, our experiments highlight that our method shines especially when a limited amount of training data is given, in terms of both pre-training and downstream MRC data.

## 2 PROPOSED METHOD

This section discusses our proposed training method called Answer-containing Sentence Generation (ASGen). While ASGen can be applied to any generative model, we use a simple Transformer (Vaswani et al., 2017) based generative model as our baseline, which we call BertGen. First, we will describe how the BertGen model generates synthetic questions and answers from a context. Next, we will explain the novel components of our methods and how we pre-trained the question generator in BertGen based on them. BertGen encodes given paragraphs with two networks, the answer generator and the question generator.

**Answer Generator.** To make the contextual embeddings and to predict answer spans for a given context without the question, we utilize a BERT (Devlin et al., 2019) encoder (Fig. 1-(1), BERT Encoder-A). We estimate the number of answer candidates $K$ by applying a fully connected layer on the contextual embedding of BERT's classification token "[CLS]". Depending on the estimated number $K$, we select the $K$ top candidate answer spans from the context. We use the $K$ selected answer spans as input to the question generator.

**Question Generator.** Next, we generate a question conditioned on each answer predicted from the answer generator. Specifically, we give as input to a BERT encoder the context and an indicator for the answer span location in the context (Fig. 1-(2), BERT Encoder-Q). Next, a Transformer

decoder generates the question word-by-word based on the encoded representation of the context and the answer span. When pre-training the question generator on an answer-containing sentence generation task, we exclude the answer-containing sentence from the original context and train the model to generate the excluded sentence given the modified context and the answer span as input.

Finally, we generate synthetic questions and answers from a large corpus, e.g., all the paragraphs in Wikipedia. After generating this data, we train the MRC model on the generated data in the first phase and then fine-tune on the downstream MRC dataset (e.g., SQuAD) in the second phase. In this paper, we use BERT as the default MRC model, since BERT or its variants achieve state-of-the-art performance across numerous MRC tasks.

## 2.1 DYNAMIC ANSWER PREDICTION

In question generation, it is important to determine which part of a given context can be a suitable answer for generating questions. To this end, we predict the number of answer $K$ in a given context $W = \{\mathbf{w}_t\}_{t=0}^{T}$ to obtain a more appropriate set of "answer-like" phrases, i.e.,

$$\{\mathbf{w}_t^{enc}\}_{t=0}^{T} = \text{BERT Encoder-A}(W),$$
$$K = \lfloor f_k(\mathbf{w}_0^{enc}) \rfloor,$$

where $T$ is the number of word tokens in the context, and position 0 reserved for classification token '$[CLS]$'. $f_k$ represents a fully connected unit with two hidden layers that have hidden dimensions equal to $H$ and 1, respectively, where $H$ is the hidden dimension of BERT Encoder-A. For training, we use the mean squared error loss between the output value of $f_k$ and ground-truth number of answers $K^{gt}$.

To calculate the score $s_i$ for start index $i$ of a predicted answer span, we compute the dot product of the encoder output with a trainable vector $\mathbf{v}_s$. For each start index $i$, we calculate the span end index score $e_{i,j}$ for index $j$ in a similar manner with a trainable vector $\mathbf{v}_e$, i.e.,

$$s_i = \mathbf{v}_s \circ \mathbf{w}_i^{enc},$$
$$e_{i,j} = \mathbf{v}_e \circ f_s(\mathbf{w}_j^{enc} \oplus \mathbf{w}_i^{enc}),$$

where $f_s$ represents a fully connected layer with hidden dimension $H$ and $\oplus$ indicates the concatenation operation. For training, we use cross-entropy loss on the $s_i$, $e_{i,j}$ and ground truth start, end of the answer span for each token. Predicting the number of answers and the answer span are jointly trained to minimize the sum of their respective losses.

During inference, we choose the $K$ top answer spans with the highest score summation of start index score and end index score, i.e.,

$$A^{span} = \{(i,j) \mid 1 \le i < T \text{ and } i \le j < T\},$$
$$a_k = \max(\{a \mid \#\{(i,j) \mid (i,j) \in A^{span} \text{ and } s_i + e_{i,j} \ge a\} = K\}),$$
$$A_k^{span} = \{(i,j) \mid (i,j) \in A^{span} \text{ and } s_i + e_{i,j} \ge a_k\}.$$

The $K$ selected answer spans $A_k^{span}$ are then given to the question generator as input in the form of an indication of the answer span location in the given context.

## 2.2 PRE-TRAINING QUESTION GENERATOR

In order to generate questions conditioned on different answers that may arise in a context, we generate a question for each of the $K$ answers. Alberti et al. (2019) proposed a pre-training method for this generative model using the self-supervised task of generating the next-sentence. We identify several issues with this approach. This technique is not specifically conditioned on the answer, despite the answer being a strong condition for the question generation task. Also, not all sentences from a paragraph may be relevant to the questions or answers from within that paragraph, so their generation is not an ideal candidate for pre-training question generation model.

To address these issues, we modify the context to exclude the sentence containing the previously generated answer and pre-train the question generation model on the task of generating this excluded answer-containing sentence, conditioned on the answer and the modified context.

Specifically, we exclude answer-containing sentence $S^{ans}$ while retaining the answer, modifying the original context $D$ to $D^{ans}$ as

$$S^{start} = \{p \mid \text{sentence start index} = p\} \cup \{T\},$$
$$S^{ans} = \{(p_s, p_e, i, j) \mid \max(\{p{\leq}i\})_s, \min(\{p{\geq}j\})_e\},$$
$$D^{ans} = [D_{[:p_s]}; D_{[i:j]}; D_{[p_e:]}], (p_s, p_e, i, j) \in S^{ans},$$

where $(i, j) \in A_k^{span}$. Note that we change $S^{ans}$ to not exclude the answer-containing sentence for fine-tuning the question generator, i.e.,

$$S^{ans} = \{(p_s, p_e, i, j) \mid p_s = i, p_e = j\}.$$

In BertGen, we pass the previously generated answer to the generation model in the form of an additional position encoding $M^{ans}$ that indicates the answer location within the context, i.e.,

$$M^{ans} = [\mathbf{m}_0 * p_s; \mathbf{m}_1 * (j - i); \mathbf{m}_0 * (T - p_e)],$$

where $\mathbf{m}_0$ and $\mathbf{m}_1$ indicate trainable vectors corresponding to encoding id $0$ and $1$, respectively. That is, we assign the encoding id for each word in the context as $0$ and each word in the answer as $1$. $A * B$ indicates the operation of stacking vector $A$ for $B$ many times.

Next, we generate answer-containing sentence output words probability $W^o = \{\mathbf{w}_t^o\}_0^T$ as

$$C^{enc} = \text{BERT Encoder-Q}(D^{ans}, M^{ans}),$$
$$\mathbf{w}_t^g = \text{Transformer Decoder}(\{\mathbf{w}_i^g\}_{i=0}^{t-1}, C^{enc}),$$
$$\{\mathbf{w}_t^o\}_{t=0}^T = \{\text{Softmax}(\mathbf{w}_t^g E)\}_{t=0}^T,$$

where $C^{enc}$ is encoded representation of the context and $E \in \mathbb{R}^{d \times D}$ represents a word embedding matrix with vocabulary size $D$ shared between the BERT Encoder-Q and the Transformer decoder.

Finally, we calculate the loss of the generated words using the cross-entropy loss as

$$\mathbb{L} = -\left(\sum_{t=1}^T \sum_{i=1}^D \mathbf{y}_{t,i} \log(\mathbf{w}_{t,i}^o)\right)/T,$$

where $\mathbf{y}$ indicates a ground-truth one-hot vector of the answer-containing sentence word. Note that $\mathbf{y}$ is the question word in the case of fine-tuning.

In this manner, we pre-train the question generation model using a task similar to the final task of conditionally generating the question from a given answer and a context.

## 3 EXPERIMENTAL SETUP

**Pre-training Dataset.** To build the dataset for answer-containing sentence generation tasks (AS-Gen) and the synthetic MRC data for pre-training the downstream MRC models, we collect all paragraphs from the entire English Wikipedia dump and synthetically generate questions and answers on these paragraphs. We apply filtering and clean-up steps that are detailed in the appendix.

Using BertGen, we extract answers from each given paragraph, and then generate questions for each answer-paragraph pairs. Finally, we obtain 43M triples of question-answer-paragraph for the synthetic data. For pre-training on answer-containing sentence generation, we sample 25M answer-paragraph pairs (Full-Wiki) from the final Wikipedia dataset to avoid extremely short contexts less than 500 characters. For ablation studies on pre-training approaches, we sample 2.5M pairs (Small-Wiki)[1] from Full-Wiki and split 25K pairs (Test-Wiki) to evaluate the pre-training method.

**Benchmark Datasets.** In most MRC datasets, a question and a context are represented as a sequence of words, and the answer span (indices of start and end words) is annotated from the context words based on the question. Among these datasets, we choose SQuAD as the primary benchmark dataset for question generation, since it is the most popular human-annotated MRC dataset. For fair comparison with existing question generation methods, we use the same splits of SQuAD-v1.1, as

---

[1] We use the Korean Wikipedia for KorQuAD, which is 15x smaller than English Wikipedia.

Table 1: Comparison with existing question generation methods on the test set of SQuAD Split1 and Split2. Models marked as '*' indicate results we reproduced.

| Generation Model | Split1 | | | Split2 | | |
|---|---|---|---|---|---|---|
| | BLEU-4 | METEOR | ROUGE-L | BLEU-4 | METEOR | ROUGE-L |
| Du et al. (2017) | 12.3 | 16.6 | 39.8 | - | - | - |
| Zhao et al. (2018)* | 13.0 | 18.2 | 41.2 | 15.1 | 19.5 | 43.4 |
| ASs2s (Kim et al., 2019) | 16.2 | 19.9 | 44.0 | - | - | - |
| Zhao et al. (2018) | - | - | - | 16.4 | 20.3 | 44.5 |
| UniLM (Dong et al., 2019) | 22.1 | 25.1 | 51.1 | 23.8 | 25.6 | 52.0 |
| BertGen (Large) + ASGen | 22.8 | 25.3 | 51.2 | 24.6 | 25.8 | 53.0 |
| UniLM + ASGen | **23.7** | **25.9** | **52.3** | **25.3** | **26.7** | **53.3** |

Table 2: Application of ASGen to other existing question generation models. BL-4, MTR, RG-L indicate BLEU-4, METEOR, ROUGE-L.

| Test set on Split1 | BL-4 | MTR | RG-L |
|---|---|---|---|
| Zhao et al. (2018)* | 13.0 | 18.2 | 41.2 |
| + ASGen | **14.2** | **19.4** | **42.8** |
| T5 (Small)* | 15.6 | 23.3 | 37.1 |
| + ASGen | **17.0** | **24.2** | **38.9** |
| UniLM | 22.1 | 25.1 | 51.1 |
| + ASGen | **23.7** | **25.9** | **52.2** |
| Test set on Split2 | BL-4 | MTR | RG-L |
| Zhao et al. (2018)* | 15.1 | 19.5 | 43.4 |
| + ASGen | **16.4** | **20.6** | **44.7** |
| T5 (Small)* | 18.8 | 25.2 | 40.5 |
| + ASGen | **19.6** | **26.1** | **41.9** |
| UniLM | 23.8 | 25.6 | 52.0 |
| + ASGen | **25.3** | **26.7** | **53.3** |

Table 3: Comparison with existing question generation methods on the test set of MS MARCO and NewsQA. (L) indicate (Large).

| MS MARCO | BL-4 | MTR | RG-L |
|---|---|---|---|
| Zhao et al. (2018) | 17.2 | - | - |
| Tuan et al. (2020) | 18.3 | 19.4 | 42.8 |
| Ma et al. (2020) | 20.5 | 24.7 | 49.9 |
| BertGen (L) + ASGen | **22.9** | **26.7** | **51.8** |
| NewsQA | BL-4 | MTR | RG-L |
| Zhou et al. (2017) | 9.9 | 16.7 | 42.3 |
| Liu et al. (2019) | 11.1 | 17.4 | 43.2 |
| Tuan et al. (2020) | 12.4 | **19.0** | 44.1 |
| BertGen (L) + ASGen | **13.8** | 18.6 | **44.5** |

previously done in Du et al. (2017), Kim et al. (2019), and Dong et al. (2019). We refer to this dataset as Split1. This split has 77K/10K/10K samples for train/dev/test sets. We also evaluate on the reversed dev-test split, referred to as Split2.[2] Additionally, we test our question generation on MS MARCO (Nguyen et al., 2016) and NewsQA (Trischler et al., 2017) for evaluating generalization of our method to other datasets. In the case of MS MARCO, questions are collected from real user query logs in Bing. For these datasets, we follow pre-processing of Tuan et al. (2020), sampling a subset of original data where the answers are sub-spans of their corresponding paragraphs to obtain train/dev/test sets with 51K/6K/7K samples for MS MARCO and 76K/4K/4K samples for NewsQA. To calculate the scores BLEU-4 (Papineni et al., 2002a), METEOR (Banerjee & Lavie, 2005b), and ROUGE-L (Lin, 2004), we use the scripts from Du et al. (2017).

To evaluate the effectiveness of generated synthetic MRC data, we test the fine-tuned MRC model on the downstream MRC dataset after training on the generated synthetic data. We calculate the EM/F1 score of the MRC model on SQuAD-v1.1 and v2.0 development set. We also evaluate on the test set of KorQuAD, a Korean dataset created with the same procedure as SQuAD-v1.1.

To further demonstrate the effectiveness of our approach, we additionally conduct experiments on question generation with Natural Questions (Kwiatkowski et al., 2019) and on the downstream MRC task with QUASAR-T (Dhingra et al., 2017) and BioASQ (Tsatsaronis et al., 2015) in the appendix.

**Implementation Details.** For all experiments and models, we use all official original hyper-parameters unless otherwise stated below. For BertGen model, we use pre-trained BERT (Base and Large) as encoder and 12 stacked layers of Transformer as decoder. For large version of the model, we use 24 layers of the encoder and the decoder with 737M parameters. For dynamic answer prediction, we use the annotated answers in SQuAD for learning the number of answer candidates

---

[2]We use the same splits as provided by Du et al. (2017)

Table 4: Ablation of pre-training methods, i.e., pre-training on NS, ASGen, and ASGen without conditioning on a given answer (w/o A), on the test set of SQuAD splits. "Wiki" indicates the sentence generation score on Test-Wiki.

Table 5: Average of 10 human evaluation scores over 50 randomly picked samples from SQuAD. Each column indicates Syntax (ST), Semantics (SM), Context-Relevance (CR) and Answer-Relevance (AR) in the range 1 to 5.

| Pre-train on Small-Wiki | Wiki | Split1 | Split2 |
|---|---|---|---|
| BertGen (w/o pre-train) | - | 15.0 | 17.1 |
| BertGen+NS | 1.4 | 19.0 | 20.2 |
| BertGen+ASGen w/o A | **5.2** | 19.9 | 21.0 |
| BertGen+ASGen | **5.2** | **20.1** | **21.4** |
| Pre-train on Full-Wiki | Wiki | Split1 | Split2 |
| BertGen+NS | 3.4 | 20.6 | 22.6 |
| BertGen+ASGen | 8.2 | 22.2 | 24.2 |
| BertGen(Large)+ASGen | **8.3** | **22.8** | **24.6** |

| Model | ST | SM | CR | AR |
|---|---|---|---|---|
| BertGen | 4.04 | 3.93 | 4.20 | 3.25 |
| BertGen+NS | 4.60 | 4.54 | 4.49 | 3.63 |
| BertGen+ASGen | **4.71** | **4.69** | **4.74** | **4.14** |
| UniLM | 4.25 | 4.31 | 4.54 | 4.06 |
| UniLM+ASGen | **4.71** | **4.79** | **4.70** | **4.17** |

$K$ and the answer spans. For the generation of unanswerable questions in SQuAD-v2.0, we separate unanswerable and answerable cases and then train separate generation models. For all BertGen models, we pre-train the question generator for 5 epochs on Wikipedia and fine-tune it for 30 epochs on MRC dataset with batch size of 32. For other question generation models, we pre-train for 1 epoch on Wikipedia. For UniLM and T5, the input is formulated as sequence-to-sequence, the first input segment is the concatenation of context and answer, while the second output segment is a missing answer-containing sentence or a question to be generated. We use all official settings for UniLM, and use the official pre-trained weights. The training time depends on the data size and the model complexity. For Zhao et al. (2018), pre-training on Full-Wiki takes only 48 hours. Pre-training BertGen on Small-Wiki in Table 4 takes 48 hours with 8 Tesla V100 GPU, resulting in 5.1, 4.3 BLEU-4 improvement on Split1, Split2 respectively. The pre-training for BertGen (Large) with Full-Wiki takes 1,224 hours and fine-tuning takes 72 hours. For MRC models, we use BERT (Large and WWM). Mecab (Kudo, 2006) is used for Korean tokenizer.

**Comparison of the Pre-training Method.** We compare ASGen with a method from Alberti et al. (2019), which is pre-training on next-sentence generation task (NS), and with a method from Golub et al. (2017), which only trains the generative model on the final MRC dataset. We reproduced these methods on BertGen as described in their original work and evaluate question generation scores on the SQuAD splits as well as corresponding sentence generation scores on Test-Wiki.

**Comparison of Downstream Results.** To check the effectiveness of our method on downstream MRC tasks, we evaluate our generated synthetic data on SQuAD-v1.1, v2.0, and KorQuAD by training MRC models (BERT and BERT+CLKT) on generated data followed by fine-tuning on the train set for each dataset. The structure of BERT+CLKT model is the same as that of original BERT except that the model is pre-trained for the Korean language. Due to the absence of common pre-trained BERT for Korean, we used this model as a baseline.

# 4 Experimental Results

## 4.1 Question and Answer Generation

**Comparison to Existing Methods.** To evaluate ASGen, we fine-tune the question generation models on both SQuAD splits, after pre-training on answer-containing sentence generation task. As shown in Table 1, 'BertGen (Large) + ASGen' and 'UniLM + ASGen' outperforms UniLM on both splits. As shown in Table 3, 'BertGen (Large) + ASGen' outperforms all existing models on all scores on both MS MARCO and NewsQA, except for comparable METEOR scores in NewsQA.

**Application to Existing Methods.** As shown in Table 2, ASGen consistently improves the performance when applied to other question generation models such as Zhao et al. (2018), T5 (Small), and UniLM across all metrics for both splits. In particular, applying ASGen on UniLM further improves its question generation capability, achieving BLEU-4, METEOR, and ROUGE-L as 23.7, 25.9, 52.2, and 25.3, 26.7, 53.3 on both splits, respectively. We reproduce Zhao et al. (2018) and T5, and use the official code of UniLM with no architecture or parameter changes.

**Ablation Study of Pre-training Task.** We also compare the BLEU-4 scores between various pre-training tasks to show the effectiveness of ASGen. As shown in Table 4, ASGen outperforms NS in the recreation score of sentence on Test-Wiki, e.g. 5.2 vs. 1.4 in Small-Wiki and 8.2 vs. 3.4 in Full-Wiki. Also, ASGen outperforms NS in question generation, e.g. 22.2 vs. 20.6 and 24.2 vs. 22.6 in the two splits, respectively. We also observe that conditioning on a given answer improves ASGen, e.g. 20.1 vs. 19.9 in Split1 and 21.4 vs. 21.0 in Split2.

**Human Evaluation.** Additionally, we also judge the quality of questions by human evaluation involving 10 evaluators over metrics such as syntax, validation of semantics, question to context relevance and question to answer relevance on 50 randomly chosen samples on SQuAD-v1.1 dev set. As shown in Table 5, applying ASGen consistently improves the human evaluation scores.

**Answer Prediction.** Table 6 shows the effectiveness of our method in generating the number of answers in a given context. In the case of fixed $K$, the MAE from the ground-truth is smallest at $K^{pred} = 5$ at 1.92 and 0.99 for test set of Split1 and Split2, respectively. Thresholding on the sum of the start and end logits shows an error of 2.31 and 1.12 on the two splits, respectively. In contrast, our method generates an appropriate number of answers, by reducing MAE to 1.24 and 0.76.

Table 6: Mean absolute error (MAE) between prediction $K^{pred}$ and ground-truth $K^{gt}$ on the test set of SQuAD

| Approach | MAE | |
|---|---|---|
| | Split1 | Split2 |
| Thresholding on Logits | 2.31 | 1.12 |
| Fixed-$K$ ($K^{pred} = 5$) | 1.92 | 0.99 |
| Dynamic-$K$ (ASGen) | **1.24** | **0.76** |

## 4.2 DOWNSTREAM MRC TASK PERFORMANCE

To show the effectiveness of the generated synthetic data, we train MRC models on generated data, before fine-tuning on the downstream data. As shown in Table 7, the synthetic data generated by 'BertGen (Large) + ASGen' consistently improves the performance of BERT (Large, WWM) by a significant margin. Pre-training BERT on synthetic data improves F1 scores by 1.8 on SQuAD-v1.1 and 5.6 on SQuAD-v2.0 for BERT (Large), and 0.7 on SQuAD-v1.1 and 2.5 on SQuAD-v2.0 for BERT (WWM). Synthetic data also improves BERT+CLKT performance on KorQuAD. Also, to show improvement due to our pre-training method in the downstream MRC task, we compare between the EM/F1 scores of BERT (Large) models trained on synthetic data generated by different question generation models, 'BertGen', 'BertGen + NS' and 'BertGen + ASGen'. As shown in Table 8, our method outperforms other methods both on SQuAD-v1.1 and SQuAD-v2.0.

## 4.3 EFFECTS OF DOWNSTREAM AND SYNTHETIC DATA SIZE

Fig. 2 shows the effects of varying amounts of downstream MRC data and synthetic data on F1 scores of BERT (Large). In Fig. 2-(a), where we fix the size of synthetic data as 43M, pre-training with 'BertGen + ASGen' consistently outperforms 'BertGen + NS' for all sizes of downstream data. While the performance difference is particularly apparent for smaller sizes of downstream data, it still persists even on using the entire MRC data (SQuAD-v1.1). In Fig. 2-(b), we also conduct experiments by training BERT (Large) using different amounts of generated synthetic data, while using the full size of downstream MRC data. The total number of pre-training steps for all data sizes is kept the same as that of 10M synthetic data. Increasing the amount of synthetic data used consistently improves the accuracy of the MRC model.

## 4.4 QUALITATIVE ANALYSIS OF QUESTIONS GENERATION

**Comparison of Sample Questions.** We qualitatively compare the generated questions after pre-training BertGen with NS and ASGen to demonstrate the effectiveness of our method. For the correct answer "49.6%" as shown in the first sample in Table 9, the word "Fresno", which is critical to make the question specific, is omitted by NS, while ASGen's question does not suffer from this issue. Note that the word "Fresno" occurs in the answer-containing sentence. This issue also occurs in the second sample, where NS uses the word "available" rather than relevant words from the answer-containing sentence, but ASGen uses many of these words such as "most" and "popular" to generate contextually rich questions. Also, the question from NS is about "two" libraries, while the answer is about "three" libraries, showing the lack of sufficient conditioning on the answer. Similarly, the

Table 7: Comparison of downstream MRC task EM/F1 scores after pre-training on the generated synthetic data (syn data). The scores are obtained from the dev set of SQuAD-v1.1 and SQuAD-v2.0, and the dev set and the test set of KorQuAD (KQD).

| MRC | Dev-v1.1 | | Dev-v2.0 | |
|---|---|---|---|---|
| model | EM | F1 | EM | F1 |
| BERT (Large) | 83.9 | 90.9 | 78.8 | 81.8 |
| +syn data | **86.3** | **92.7** | **84.5** | **87.4** |
| BERT (WWM) | 86.5 | 92.8 | 83.1 | 85.9 |
| +syn data | **87.4** | **93.5** | **85.5** | **88.4** |
| MRC | Dev-KQD | | Test-KQD | |
| model | EM | F1 | EM | F1 |
| BERT+CLKT | 87.1 | 94.5 | 86.2 | 94.1 |
| +syn data | **87.8** | **95.0** | **86.7** | **94.6** |

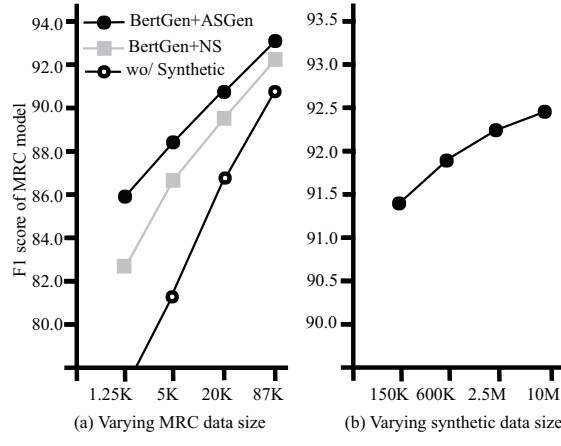

(a) Varying MRC data size    (b) Varying synthetic data size

Figure 2: F1 scores of BERT (Large) on SQuAD-v1.1 dev by limiting size of MRC and synthetic data.

Table 8: Comparison of downstream MRC task EM/F1 scores using the synthetic data from different pre-training methods. The scores are obtained from SQuAD-v1.1 and SQuAD-v2.0 dev set.

| MRC model | Synthetic Data generated by | SQuAD-v1.1 | | SQuAD-v2.0 | |
|---|---|---|---|---|---|
| | | EM | F1 | EM | F1 |
| BERT(Large) | BertGen (w/o pre-train) | 85.1 | 91.4 | 80.9 | 83.9 |
| | BertGen+NS | 85.6 | 92.3 | 81.5 | 85.8 |
| | BertGen+ASGen | **86.3** | **92.7** | **84.5** | **87.4** |

third example also shows that ASGen generates more contextual questions than NS by including the exact subject "TARDIS" based on the corresponding answer. Based on these observations and from the score improvements in Table 4, we conjecture that ASGen leads the question generation model to better condition on the answer and to generate more contextualized questions than NS.

**Categorization of Reasoning Type.** We manually categorized the reasoning type of 150 randomly sampled generated questions on Wikipedia for both answerable and unanswerable questions. The results Table 10 and Table 11 show that generated questions using ASGen often require multi-hop or other non-trivial reasoning. We follow the same categorization as done by SQuAD-v1.1 (Rajpurkar et al., 2016) and SQuAD-v2.0 (Rajpurkar et al., 2018). Note that each example can be assigned to multiple reasoning types for the answerable questions.

## 5 RELATED WORK

**Question Generation.** Research on question generation has a long history, such as Kalady et al. (2010) and Skalban et al. (2012). Researchers have actively studied question generation for various purposes, including for data augmentation in question answering. Du et al. (2017) proposed an attention-based model for question generation by encoding sentence-level as well as paragraph-level information. Zhao et al. (2018) utilized a gated self-attention encoder with a max-out unit to handle long paragraphs. Song et al. (2018) introduced a query-based generative model to jointly solve question generation and answering tasks. Kim et al. (2019) separately encoded the answer and the rest of the paragraph for question generation. Ma et al. (2020) suggested sentence-level semantic matching and answer-position-aware question generation. Tuan et al. (2020) show that incorporating interactions across multiple sentences enhances question generation performance. Our approach can further improve the question generation quality of these methods by pre-training them with the answer-containing sentence generation task.

**Transfer Learning.** Pre-training methods are popular in natural language processing for learning contextualized word representations. Open-GPT (Radford et al., 2018), BERT (Devlin et al., 2019), XLNet (Yang et al., 2019), PEGASUS (Zhang et al., 2019), ERNIE-GEN (Xiao et al., 2020), UniLM

Table 9: Examples of questions generated on SQuAD-v1.1 development set. We compare generated questions from 'BertGen + ASGen' with 'BertGen + NS'. Colored Text indicates given answers.

| | |
|---|---|
| Context | (omit) ... The population density was 4,404.5 people per square mile. (1,700.6km). The racial makeup of Fresno was 245,306 ( 49.6% ) White, 40,960 (8.3%) ... (omit) |
| BertGen + NS | What percent of the population is White? |
| BertGen + ASGen | What percentage of the Fresno population is White? |
| Context | (omit) ... in the world. Cabot Science Library, Lamont Library, and Widener Library are three of the most popular libraries for undergraduates to use ... (omit) |
| BertGen + NS | Which two libraries are available for undergraduates to use? |
| BertGen + ASGen | What are the three most popular libraries for undergraduates? |
| Context | (omit) ... in a stolen Mark I Type TARDIS "Time and Relative Dimension in Space" time machine which allows him to travel across time and space. ... (omit) |
| BertGen + NS | What does the doctor refer to? |
| BertGen + ASGen | What does the TARDIS stand for? |

Table 10: Manual categorization of the reasoning type for 150 randomly sampled answerable questions generated questions on Wikipedia. Note that each example can be assigned to multiple types.

| Reasoning Type | BertGen +ASGen | SQuAD v1.1 |
|---|---|---|
| Lexical Variation (Synonymy) | 40.7% | 33.3% |
| Lexical Variation (World Knowledge) | 4.0% | 9.1% |
| Syntactic Variation | 53.3% | 64.1% |
| Multi Sentence Reasoning | 21.3% | 13.6% |
| Ambiguous/Unanswerable | 4.0% | 6.1% |

Table 11: Manual categorization of the reasoning type for unanswerable questions.

| Reasoning Type | BertGen +ASGen | SQuAD v2.0 |
|---|---|---|
| Negation | 8.0% | 9.0% |
| Antonym | 14.7% | 20.0% |
| Entity Swap | 36.0% | 21.0% |
| Mutual Exclusion | 9.3% | 15.0% |
| Impossible Condition | 7.3% | 4.0% |
| Other Neutral | 19.3% | 24.0% |
| Answerable | 5.3% | 7.0% |

(Dong et al., 2019), UniLMv2 (Bao et al., 2020), T5 (Raffel et al., 2020) and BART (Lewis et al., 2020) utilize Transformer (Vaswani et al., 2017) to learn different types of language models on a large dataset followed by fine-tuning on a downstream task. These pre-training approaches tend to be very generic, while our approach is a more appropriate pre-training method focused on the specific task of question generation. Lee et al. (2019b) suggested a pre-training method for information retrieval called Inverse Cloze Task which treats a sentence as a pseudo-query and its surrounding context as a pseudo-target. Unlike this method, our pre-training task for the question generator is strongly conditioned on the answer and focuses on generating missing answer-containing sentence in the context to learn better representations more suitable to the question generation task.

**Synthetic Data Generation.** Subramanian et al. (2018) show that neural models generate better candidate answers from a given paragraph than using off-the-shelf tools for selecting named entities and noun phrases. Yang et al. (2017) introduced a training method for the MRC model by combining synthetic data and human-annotated data. Similar to our method, Golub et al. (2017) proposed to generate questions conditioned on generated answers by separating the answer generation and the question generation. Unlike this paper, they do not estimate the number of answers, and they do not pre-train their question generator. Dong et al. (2019) also show that utilizing synthetic data boosts the performance of MRC models. Inspired by these previous studies, we propose a newly designed pre-training technique that improves capability of question generation models.

## 6  Conclusions

We propose a novel pre-training method called ASGen to learn generating contextually rich questions better conditioned on the answers. Our approach improves question generation ability of existing methods, achieves new state-of-the-art results on MS MARCO and NewsQA, and the synthetic data increases downstream MRC accuracy across a wide range of datasets, such as SQuAD-v1.1, v2.0, and KorQuAD, without any modification to the existing MRC models.

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

## APPENDIX

## A    QUESTION GENERATION ON MORE MRC DATASETS

We also evaluate the question generation model on another data split (Split3) from Zhao et al. (2018). Split3 is obtained by dividing the original development set in SQuAD-v1.1 into two equal halves randomly and choosing one of them as the development set and the other as test set while retaining the train set in SQuAD-v1.1. As shown in Table 12, applying ASGen to the reproduced question generation model from Zhao et al. (2018) improves BLEU-4, METEOR, and ROUGE-L score on Split3 by 1.3, 0.9, and 1.3, respectively.

We also test generalization capability of our method by evaluating on Natural Questions (Kwiatkowski et al., 2019) dataset, where questions are collected from real user query logs on Google and may have less biased questions than other datasets. As shown in Table 13, applying ASGen to BertGen shows improvement in the question generation score on Natural Questions short answer dataset by 3.8 BLEU-4, 2.5 METEOR and 1.1 ROUGE-L.

Table 12: Additional experiments on the effectiveness of ASGen on the test set of SQuAD Split3. Small-Wiki is used to pre-train the models. Models with '*' indicate those results we reproduced.

| Model + pre-training method | BLEU-4 | METEOR | ROUGE-L |
|---|---|---|---|
| Zhao et al. (2018) | 16.8 | 20.6 | 44.9 |
| Zhao et al. (2018)* | 16.3 | 20.3 | 44.5 |
| Zhao et al. (2018)* + ASGen | **17.6** | **21.2** | **45.8** |

Table 13: Ablation study of applying ASGen to question generation model on Natural Questions (Kwiatkowski et al., 2019) short answer dataset. The scores are obtained from the dev set.

| Model + pre-training method | BLEU-4 | METEOR | ROUGE-L |
|---|---|---|---|
| BertGen (Large) | 31.5 | 30.4 | 60.2 |
| BertGen (Large) + ASGen | **35.3** | **32.9** | **61.3** |

## B    TRAINING ELECTRA MRC MODEL WITH GENERATED SYNTHETIC DATA

We also apply our synthetic data generated from Small-Wiki to another MRC model, ELECTRA (Clark et al., 2020), which shows the state-of-the-art results. In Table 14, we report the mean EM/F1 score on SQuAD 2.0 development set of four runs by using official Electra source code[3] and the pre-trained checkpoint. Pre-training ELECTRA on the generated synthetic data using ASGen improves 0.5 EM and 0.6 F1 score on the downstream MRC dataset, SQuAD-v2.0, even when using only Small-Wiki.

---

[3]https://github.com/google-research/electra

Table 14: Ablation study of applying our method to ELECTRA (Clark et al., 2020) on SQuAD-v2.0 dev set after pre-training on the generated synthetic data using ASGen with Small-Wiki.

| MRC model | Synthetic Data | Dev set | |
|---|---|---|---|
| | | EM | F1 |
| ELECTRA | - | 87.4 | 90.2 |
| (Large) | 'Small-Wiki' | **87.9** | **90.8** |

## C  Transfer Learning to Other MRC Dataset (QUASAR-T)

To show that our generated data is useful for other MRC datasets, we fine-tune and test the MRC model on QUASAR-T (Dhingra et al., 2017), which is another large-scale MRC dataset, after training on the synthetic data generated from SQuAD-v1.1. In this experiment, we first fine-tune 'Bert-Gen + ASGen' using SQuAD-v1.1, and using synthetic data generated by this model, we train the BERT (Large) MRC model. Afterwards, we fine-tune BERT (Large) for the downstream MRC task using QUASAR-T data. QUASAR-T has two separate datasets, one with short snippets as context, and the other with long paragraphs as context. As shown in Table 15, training with our synthetic data improves the F1 score on the test set by 2.2 and 1.7 for the two cases, respectively.

Table 15: EM/F1 scores of the BERT (Large) fine-tuned on QUASAR-T dataset. The used synthetic data is generated from ASGen trained on SQuAD-v1.1 (Full-Wiki).

| MRC model | Synthetic Data | Short(Dev) | | Short(Test) | |
|---|---|---|---|---|---|
| | | EM | F1 | EM | F1 |
| BERT | - | 74.3 | 78.6 | 74.1 | 77.8 |
| | Full-Wiki | **76.5** | **80.1** | **76.5** | **80.0** |
| MRC model | Synthetic Data | Long(Dev) | | Long(Test) | |
| | | EM | F1 | EM | F1 |
| BERT | - | 72.1 | 75.6 | 72.1 | 74.8 |
| | Full-Wiki | **74.2** | **77.4** | **73.8** | **76.5** |

## D  Comparison of Answer Generation Approaches on MRC task

We also evaluate the effectiveness of dynamic-$K$ answer prediction by pre-training the BERT (Large) (Devlin et al., 2019) MRC model on our synthetic data from Small-Wiki followed by fine-tuning on the downstream MRC dataset, SQuAD-v2.0. As shown in Table 16, dynamic-$K$ answer prediction shows 0.3 EM and 0.2 F1 score improvements from the baseline approach, fixed-$K$.

Table 16: Comparison of predicting $K$ answers with downstream BERT (Large) MRC results on SQuAD-v2.0 dev set after pre-training on each generated synthetic data using corresponding answer generation approach with Small-Wiki.

| Answer generation approach | Dev set | |
|---|---|---|
| | EM | F1 |
| Fixed-$K$ ($K^{pred} = 5$) | 81.38 ($\pm$0.09) | 84.36 ($\pm$0.07) |
| Dynamic-$K$ (ASGen) | **81.73** ($\pm$0.06) | **84.62** ($\pm$0.04) |

## E  Details of Wikipedia Preprocessing

To build the answer-containing sentence generation data and the synthetic MRC data for SQuAD (Rajpurkar et al., 2016), we collect all paragraphs from all articles of the entire English Wikipedia dump and generate questions and answers on these paragraphs. We apply extensive filtering and clean-up to only retain the highest-quality paragraphs from Wikipedia, as follows.

To filter out low-quality articles, we remove those with less than 200 cumulative page-views including all re-directions in a two-month period. In order to calculate the number of page-views, official Wikipedia page-view dumps were used. Of the 5.4M original Wikipedia articles, filtering by page-views leaves 2.8M articles. We also remove those articles with less than 500 characters, as they are often low-quality stub articles, which further removes additional 16% of the articles. We remove all "meta" namespace pages such as talk, disambiguation, user pages, portals, etc. as they often contain irrelevant text or casual conversations between editors. In order to extract clean text from the wiki-markup format of the Wikipedia articles, we remove extraneous entities from the markup including table of contents, headers, footers, links/URLs, image captions, IPA double parentheticals, category tables, math equations, unit conversions, HTML escape codes, section headings, double brace templates such as info-boxes, image galleries, HTML tags, HTML comments, and all tables.

We then split the cleaned text into paragraphs and remove all paragraphs with less than 150 characters or more than 3,500 characters. Paragraphs with the number of characters between 150 to 500 were sub-sampled such that these paragraphs make up 16.5% of the final dataset, as originally done for the SQuAD dataset. Since the majority of the paragraphs in Wikipedia are rather short, out of the 60M paragraphs from the final 2.4M articles, our final Wikipedia dataset contains 8.3M paragraphs. Finally, we generate 43M answer-paragraph pairs from the final Wikipedia dataset with the answer generator of BertGen in this paper.

## F    TRANSFER LEARNING TO OTHER LIMITED DOMAIN DATA (BIOASQ)

We conduct experiments on BioASQ (Tsatsaronis et al., 2015) dataset to show the effectiveness of our model in limited-data domains having less annotated data. As shown in Table 17, ASGen improves the question generation scores by 6.0 BLEU-4, 7.8 METEOR and 6.9 ROUGE-L on BioASQ factoid-type 6b. Moreover, using 'Full-Wiki' data enhances the performance of BERT(Large) by a large margin and outperforms BioBERT (Lee et al., 2019a), by 0.95 Macro F1 (Yes/No) and 1.63 F1 (List). Note that BioBERT is specifically pre-trained on a medical corpus (PubMed) whereas we use a generic corpus Wikipedia, 'Full-Wiki', with our generation models fine-tuned on SQuAD.

Table 17: The performance of our method on limited-data domain (BioASQ). Note that the scores of question generation are obtained from BioASQ factoid-type 6b. All experiments were conducted using the official source code of Yoon et al. (2020).

| Question Generation Model | | BLEU-4 | METEOR | ROUGE-L |
|---|---|---|---|---|
| BertGen (Large) | | 6.6 | 10.0 | 33.1 |
| BertGen (Large) + ASGen (Full-Wiki) | | **12.6** | **17.8** | **40.0** |
| MRC model | Pre-training Data | Factoid (MRR) | Yes/No (Macro F1) | List-Type (F1) |
| BERT(Large) | - | 34.3 | 53.8 | 36.1 |
| BERT(Large) | ASGen (Full-Wiki) | 49.2 | **81.1** | **39.8** |
| BioBERT(Large) | PubMed | **52.3** | 80.1 | 38.1 |

## G    APPLICATION OF ASGEN TO T5 WITH LIMITED PRE-TRAINING DATA

As shown in Table 18, our pre-training method, ASGen, increases question generation scores of T5 (Small) (Raffel et al., 2020) model even using limited pre-training data of Small-Wiki. We expect our pre-training may show a similar effect in other sized T5 models as well. Results for T5 pre-training with Full-Wiki data are in the main paper.

## H    CENTRAL TENDENCY AND VARIATION FOR HUMAN EVALUATION

Human evaluation involves 10 evaluators over metrics such as syntax (ST), validation of semantics (SM), question to context relevance (CR) and question to answer relevance (AR) on 50 randomly chosen samples on SQuAD-v1.1 development set. Each score is in the range 1 to 5. Central tendency and variation can be found in Table 19.

Table 18: Application of ASGen to T5 Model with Limited Pre-Training data

| Test set on Split1 | BLEU-4 | METEOR | ROUGE-L |
|---|---|---|---|
| T5 (Small) | 15.6 | 23.3 | 37.1 |
| T5 (Small) + ASGen (Small-Wiki) | **16.5** | **24.0** | **38.4** |
| Test set on Split2 | BLEU-4 | METEOR | ROUGE-L |
| T5 (Small) | 18.8 | 25.2 | 40.5 |
| T5 (Small) + ASGen (Small-Wiki) | **19.2** | **25.9** | **41.3** |

Table 19: Central tendency and variation for human evaluation scores. $\pm$ is 95% confidence interval.

| Model | ST | SM | CR | AR |
|---|---|---|---|---|
| BertGen | 4.04 $\pm$0.18 | 3.93 $\pm$0.19 | 4.20 $\pm$0.16 | 3.25 $\pm$0.22 |
| BertGen + NS | 4.60 $\pm$0.12 | 4.54 $\pm$0.13 | 4.49 $\pm$0.14 | 3.63 $\pm$0.22 |
| BertGen + ASGen | **4.71** $\pm$0.10 | **4.69** $\pm$0.11 | **4.74** $\pm$0.09 | **4.14** $\pm$0.18 |
| UniLM | 4.25 $\pm$0.16 | 4.31 $\pm$0.16 | 4.54 $\pm$0.12 | 4.06 $\pm$0.19 |
| UniLM + ASGen | **4.71** $\pm$0.11 | **4.79** $\pm$0.09 | **4.70** $\pm$0.11 | **4.17** $\pm$0.18 |

## I  CENTRAL TENDENCY AND VARIATION FOR THE DOWNSTREAM TASKS

For the EM and F1 scores on downstream SQuAD-v1.1 and v2.0 development set in Table 7 of our main paper, we selected 5 model checkpoints from the same pre-training on the synthetic data in different numbers of training steps. We then fine-tuned each of these models on the final downstream data three times each, chose the best performing model on the development set, and reported its score. Central tendency and variation can be found in Table 20.

Table 20: Central tendency and variation for the score of our approach, BertGen(Large) + ASGen, on downstream SQuAD-v1.1 and v2.0 dataset. $\pm$ is standard deviation.

| MRC model | Synthetic Data | Dev-v1.1 | | Dev-v2.0 | |
|---|---|---|---|---|---|
| | | EM | F1 | EM | F1 |
| BERT (Large) | Full-Wiki | 86.2 $\pm$0.1 | 92.7 $\pm$0.1 | 84.4 $\pm$0.2 | 87.3 $\pm$0.1 |
| BERT (WWM) | Full-Wiki | **87.4** $\pm$0.1 | **93.4** $\pm$0.1 | **85.5** $\pm$0.1 | **88.3** $\pm$0.1 |

## J  DETAILS OF GENERATING UNANSWERABLE QUESTIONS

The mechanism of generating questions may differ in generating answerable and unanswerable questions. For example, the model could exploit a mismatching phrase to make a question plausible but unanswerable. In order to reflect these characteristics, we train answerable and unanswerable models separately. We first take the BertGen model pre-trained on the ASGen task and then fine-tune this model on the no-answer question generation on SQuAD-v2.0. We infer with this model on the entire Wikipedia to make negative examples for un-answerble synthetic data for pre-training MRC models on SQuAD-v2.0.

## K  DISCUSSION ON WEAK SUPERVISION FOR DYNAMIC-$K$ PREDICTION

In question generation, it is important to find which elements of a given context are suitable answer. To do this, we predict the number of answers to obtain a more appropriate set of "answer-like"

phrases that humans tend to choose when they are preparing a question, rather than all possible entity phrases. This tendency can also be found in the SQuAD dataset, which has a varying number of annotated answers per context, even though the annotators were recommended to create up to five answers, as shown in Table 21. While we do not have the ground-truth number of answers for all contexts, this characteristic of SQuAD annotation can still be a useful weak supervision for learning the number of answer candidates.

Table 21: Distribution over the number of answers in SQuAD-v1.1 dataset.

| Number of Answers | 1 | 2 | 3 | 4 | 5 | 6+ |
|---|---|---|---|---|---|---|
| Percentage of Sample | 0.5 | 0.9 | 9.1 | 21.9 | 60.1 | 7.5 |

## L    BLEU-4, METEOR, AND ROGUE-L

BLEU (Papineni et al., 2002b), METEOR (Banerjee & Lavie, 2005a) and ROUGE (Lin, 2004) are widely-used metrics for evaluating the quality of generated text, where the quality indicates the degree of correspondence between generated text and reference texts. BLEU uses modified precision to compare a generated text against the reference texts. BLEU-4 calculates a weighted score of unigram, bigram, trigram, and 4-gram based matching. METEOR uses harmonic mean between precision and recall of unigrams, but with for recall given more importance than precision. Unlike BLEU, METEOR also tries to match synonyms and performs stemming instead of just relying on exact word matching. ROUGE-L is the longest common sub-sequence based word matching. The longest co-occurrence in sequences of n-grams between generated text and reference texts are considered for calculating the score. To calculate these evaluation scores, we follow the script from Du et al. (2017), except for the corresponding scripts from other question generation models when ASGen is applied to them.

## M    LINKS TO DOWNLOADABLE COMPONENTS

For Wikipedia data, we downloaded English Wikipedia dump in Feb 2019 from (`https://dumps.wikimedia.org/enwiki/latest/enwiki-latest-pages-articles.xml.bz2`). Page views were obtained from (`https://dumps.wikimedia.org/other/pageviews/2019/2019-01/`) and (`https://dumps.wikimedia.org/other/pageviews/2019/2019-02/`). For applying our method to other existing question generation models, we reproduce Zhao et al. (2018) using publicly available code (`https://github.com/seanie12/neural-question-generation`), Raffel et al. (2020) using publicly available code (`https://github.com/patil-suraj/question_generation`) and use the official code of Dong et al. (2019) (`https://github.com/microsoft/unilm`).

