# OpenReview forum: "Learning to Generate Questions by Recovering Answer-containing Sentences"
_ICLR.cc/2021/Conference — Reject_

### Official Review · AnonReviewer3 · 2020-10-28
**A new method to pre-train question generation systems for MRC data augmentation which leads to consistent improvements. The initial selection step needs to be exposed better but the findings are otherwise supported by thorough experiments.**

**Rating:** 7
**Confidence:** 4

**Review:**

### Summary

This work presents a new multi-step method to pre-train a question generation system, which can then be used to create synthetic data to improve a Machine Reading Comprehension system.

First, the authors train a system to identify spans in a text paragraphs which would constitute likely answers for questions about the paragraph. Then, they pre-train a system to generate those questions by taking the selected answer and surrounding sentences as input, and generating the sentence which contained the answer. Finally, they use real MRC data to fine-tune the question generation system.

The authors first provide a direct evaluation of their method by showing a consistent improvement in reference-based metrics comparing to gold questions for a given paragraph answer, then shows that the generated synthetic data also leads to improvement on the downstream MRC task for the SQuAD dataset, especially when using less annotated data.

The results provided are encouraging and accompanied by a number of enlightening supporting experiments. The paper could still be improved by clarifying specific points.

### Clarity

While the experimental setting description gives a broad idea of what the authors did, some details are missing for full reproducibility. Most notably, there is very little information on the first step setup, training the answer selection system. The authors also describe their setting based on BERT but not the one with UniLM

What is the dimension of the indicator vector that tells the model where the answer is located? Is there a particular reason why the authors decided to use an indicator vector rather than e.g. use a [SEP] token?

Dynamic prediction of the number of answers in a paragraph doesn't seem to account for much improvement. Can the authors provide some measure of statistical significance? Otherwise, the method really is novel enough without that part.

Did the model evaluate the effect of using synthetic data on the other tasks?

### Correctness

The claims are mostly well supported by the experiments (to the exception of the dynamic K prediction).

##### Additional citations
The pre-training sentence prediction method is related to:
**Latent retrieval for weakly supervised open domain question answering**, Lee et al., *ACL 2019*

---

> ### Author Response · Authors · 2020-11-17
> **Response to AnonReviewer3**
>
> **1. Regarding detailed explanation of the answer-generation system and utilizing UniLM**
>
> We use the standard pre-trained BERT to initialize the answer generation model, which is then further finetuned on SQuAD. For UniLM, as discussed in *Section4->Question and Answer Generation->Application to Existing Methods*, we used the official question generation source code of UniLM (Dong et al., 2019) with no architecture changes and using all official hyperparameters. We used the official pre-trained UniLM checkpoint and further pre-trained it on the task of generating answer-containing sentences, before finetuning on SQuAD for question generation. As mentioned in *Section 3->Pre-training Dataset*, all the pre-training data and synthetic data are created by BertGen. We will clarify this in our manuscript.
>
> **2. Regarding the dimensions of the indicator vector and the reason for using this in BertGen**
>
> The indicator vector is a trainable vector and is utilized to indicate directly where the answer is located in the context. The dimension of each indicator vector is the same as the hidden embedding size of the BertGen model. We did not attempt to use the [SEP] token to indicate the answer, but we think it is an interesting research direction to determine which type of answer indication works best.
>
> **3. Statistical significance of dynamic prediction of the number of answers**
>
> In *Table 13*, we show the downstream effect of dynamic K even on using Small Wiki data, and we expect the difference to be larger on Full Wiki. We provide the standard error of our mean estimates for Fixed-K and Dynamic-K as below -
>
> | Answer Generation Approach    |       EM                     |       F1      |
> |----------------------------|:-------------|:-------------|
> | Fixed-K (K=5)              | 81.38(±0.09)  | 84.36(±0.07)  |
> | Dynamic-K (Ours)           | **81.73(±0.06)**  | **84.62(±0.04)**  |
>
> **4. Question about evaluating the effect of using synthetic data on other tasks**
>
> We provide the effect of using synthetic data on SQuAD, KorQuAD, and QUASAR-T datasets. If the reviewer meant non-QA tasks by “other tasks”, we did not evaluate the efficacy of ASGen synthetic data in non-QA tasks, though this would be an interesting research direction. If the review could suggest particular potential tasks, we will seriously consider them.
>
> **5. For missing related work**
>
> We will update the missing references in our manuscript.
>
> **References**
>
> Dong et al. (2019) : Unified language model pre-training for natural language understanding and generation. (NeurIPS 2019)

---

### Official Review · AnonReviewer1 · 2020-10-28
**Method makes sense, but needs more experiments**

**Rating:** 5
**Confidence:** 4

**Review:**

The submission introduces a new approach for pre-training models for question generation conditioned on an answer and a document. Given a document, a set of k answers is chosen by a classifier. Then, the sentence containing the answer is removed, and a sequence to sequence model is used to reconstruct it. This pre-training task more closely matches the desired end task. Results show that this method improves question generation performance compared to previous work, and that the synthetic questions can be used to improve question answering models. Overall, I think the approach makes a lot of sense. However, I think it needs to compare with stronger pre-training baselines, and needs a more thorough comparison of the effect on question answering performance.

The paper primarily evaluates on question generation for a range of datasets, and shows improvements on both automated and human evaluation metrics. I think it's fair to say that the question generation is quite a niche task, perhaps because of limited applications. Indeed, the abstract of the paper pitches the task as being primarily useful for generating synthetic data for question answering. That's fine, but given this motivation, I think the evaluation should focus much more on the downstream impact on question answering.

I'd really like to see if the method still improves question generation performance on top of more recent sequence to sequence models, such as T5 or BART, which have both been available for almost a year. As far as I know, these both significantly outperform UniLM on all the comparisons I've seen. The pre-training objectives used there, which involve predicting masked spans, seem closely related to the proposed method. This experiment would help understand whether the proposed method adds anything on top of more general pre-training approaches.

The paper convincingly demonstrates that synthetic questions improve a baseline BERT question answering model, which was already shown by e.g. Alberti et al (2019). However, I don't think the paper does much to suggest that the synthetic questions from the proposed method are better for QA than other approaches - for example, Alberti et al. appear to report similar numbers with their method. It's important to include a baseline set of synthetic questions, as it's not at all clear that improvements on question generation metrics will correlate with usefulness for QA training.

Again, the question answering results would be more convincing if they hold up with more recent models than BERT, which achieve better scores without synthetic questions. Table 11 does give a result using Electra - but gives the baseline Electra model an exact match score of 87.4. The Electra paper instead lists 88.0, slightly higher than the results using synthetic questions.

---

> ### Author Response · Authors · 2020-11-17
> **Response to AnonReviewer1**
>
> **1. Regarding the impact on downstream question answering**
>
> We have provided the downstream results on SQuAD 1.1, SQuAD 2.0, KorQuAD (*Table 7*), and QUASAR-T long and short (*Table 11*) to demonstrate that our approach indeed helps in improving the prediction accuracy of the main downstream MRC tasks. Furthermore, we have also provided final downstream MRC scores for limited MRC data as well as for low pre-training data settings (*Figure 2*).
>
> **2. Regarding the comparison of applying ASGen to T5 (Raffel et al., 2020) or BART (Lewis et al., 2020), and general pre-training approaches**
>
> It is difficult to provide results of BART and T5 pre-trained with ASGen within the limited rebuttal period due to the large compute requirements for pre-training of these models.
> Conventional pre-training approaches such as that of BART tend to be very generic, while our approach is a more appropriate pre-training method focused on the specific task of question generation, with strong conditioning on the answer. We have compared our pre-training method to that of Alberti et al. (2019) and UniLM (Dong et al. 2019), and we show that our focused pre-training is indeed more effective than these generic pre-training methods for question generation.
>
> **3. Comparison of synthetic data from different pre-training methods on downstream MRC task**
>
> The downstream MRC results of Alberti et al. (2019) as reported in their original paper on the SQuAD 2.0 dev set using BERT(wwm) is $85.1$ EM / $87.9$ F1, which is $0.4$ EM / $0.5$ F1 lower than our score in *Table 7*-BERT(wwm). Also, to show improvement of our pre-training method, we reproduced the pre-training method of Alberti et al. (2019) and generated synthetic data for downstream MRC tasks. As shown in *Figure 2* and below, using synthetic data from our method outperforms other methods using BERT(Large) on the final downstream MRC tasks -
>
> | SQuAD-v1.1  \| | Synthetic Data                                 \||   EM     \||  F1  |
> |-------------|:------------------------------------|:----|:----|
> | BERT(Large) | BertGen         | 85.1 | 91.4 |
> | BERT(Large) | BertGen + NS (Alberti et al. 2019)  | 85.6 | 92.3 |
> | BERT(Large) | BertGen + ASGen                     | **86.3** | **92.7** |
>
> | SQuAD-v2.0  \| | Synthetic Data                                 \||   EM     \||  F1  |
> |-------------|:------------------------------------|:----|:----|
> | BERT(Large) | BertGen         | 80.9 | 83.9 |
> | BERT(Large) | BertGen + NS (Alberti et al. 2019)  | 81.5 | 85.8 |
> | BERT(Large) | BertGen + ASGen                     | **84.5** | **87.4** |
>
> We will update our manuscript with these scores.
>
> **4. Regarding applying the synthetic data to Electra model (*Table 11*)**
>
> In *Table 11*, we report the mean score on SQuAD 2.0 of four runs by using official Electra source code and the pre-trained checkpoint. We found that the Electra model exhibits a large variance in results in different finetuning runs, as also mentioned in their official Github (https://github.com/google-research/electra#expected-results). We will clarify this in our manuscript.
>
> **References**
>
> Golub et al. (2017) : Two-stage synthesis networks for transfer learning in machine comprehension. (EMNLP 2017)
>
> Alberti et al. (2019) : Synthetic QA corpora generation with roundtrip consistency. (ACL 2019)
>
> Lee et al. (2019) : Latent retrieval for weakly supervised open domain question answering. (ACL 2019)
>
> Lewis et al. (2020) : BART: Denoising Sequence-to-Sequence Pre-training for Natural Language Generation, Translation, and Comprehension (ACL 2020)
>
> Raffel et al. (2020) : Exploring the limits of transfer learning with a unified text-to-text transformer. (JMLR 2020)

---

> > ### Author Response · Authors · 2020-11-24
> > **Additional Comment on "2. Regarding the comparison of applying ASGen to T5 (Raffel et al., 2020) or BART (Lewis et al., 2020), and general pre-training approaches"**
> >
> > **Results of our pre-training method on the T5 model**
> >
> > Pre-training T5 (Small) was relatively quick to train, so we are able to obtain results before the end of the rebuttal period. As shown below, our pre-training step increases question generation scores on SQuAD splits of T5 (Small) using both Small-Wiki and Full-Wiki.  We expect our pre-training may show a similar effect in other sized T5 models as well.
> >
> > | Test set on Split 1 | BLEU-4 | METEOR | ROUGE-L |
> > |---------------------------|:------:|:------:|:-------:|
> > | T5 (Small)            |   15.55  |  23.25  |   37.08  |
> > | T5 (Small) + ASGen (Small-Wiki)    |  16.46  |  23.96  |  38.39  |
> > | T5 (Small) + ASGen (Full-Wiki)    |  **16.97**  |  **24.19**  |  **38.85**  |
> >
> > | Test set on Split 2 | BLEU-4 | METEOR | ROUGE-L |
> > |---------------------------|:------:|:------:|:-------:|
> > | T5 (Small)            |   18.77  |  25.22  |   40.46  |
> > | T5 (Small) + ASGen (Small-Wiki)    |  19.18  |  25.88  |  41.29  |
> > | T5 (Small) + ASGen (Full-Wiki)    |  **19.63**  |  **26.05**  |  **41.92**  |
> >
> > The results were obtained using the source code from https://github.com/patil-suraj/question_generation with all default hyper-parameters.

---

### Official Review · AnonReviewer4 · 2020-10-30
**Solid empirical work but limited technical contributions**

**Rating:** 6
**Confidence:** 4

**Review:**

This paper proposes a pretraining technique for question-generation models. The pretraining involves generating the sentence which contains an answer candidate from the context around it. Through several well-executed experiments the paper shows that this type of pretraining can improve performance of several existing question generation models, and the resulting synthetic questions generated help in augmenting reading comprehension datasets.

Strengths:
- This is solid empirical work, with several detailed experiments showing the utility of the pretraining method on multiple benchmarks and with multiple models. It was particularly nice to see evaluation on Korean as well as English.

- The authors clearly put effort in explaining all the methods and experiments precisely and with details.

Weaknesses:
- The technical contributions of the paper are rather limited. Pretraining has been of much interest in the NLP community recently, and this paper follows a line of related works which have proposed similar techniques (e.g. Inverse Cloze Task, Lee et al, ACL 2019; T5, Raffel et al, JMLR 2020). There isn't much discussion of these existing papers either.

- It is not clear how much of the benefit of pretraining comes from the specific approach used here, versus the fact that there is some pretraining on the decoder which generates the questions. We could learn more about this if there was a comparison to other pretrained models which have a decoder, e.g. T5, BART (Lewis et al, 2020).

- In terms of the question generation task, it is well known now that Squad questions have a bias towards high lexical overlap with the passage (due to the manner in which they were constructed, see Lee et al above). This raises the question whether the approach in this paper can generalize to datasets where this bias does not exist, e.g. Natural Questions. This seems to be a limitation of not just paper, but the prior works as well.

Other comments / questions:
- Where do the ground truth answers come from during pretraining? Is the answer prediction model also pretrained?

- In section 2.1, it says that the MSE loss is computed using K, but K comes from the floor function which does not support backpropagation. Do you use the output f_k instead to compute the MSE?

- In the same section, I assume a softmax is applied on s_i, e_i,j before taking the cross-entropy loss. In this case, is e_i,j normalized over all spans in the passage or only the ones which start at i?

- Is w^o_t an embedding or a probability? There seems to be confusion at the end of section 2.

- Some more discussion of the UniLM baseline would be good for people not familiar with that work.

- Missing discussion of related work: "Yang, Zhilin, et al. "Semi-supervised qa with generative domain-adaptive nets." ACL (2017)."

- How does the model generate unanswerable questions for Squad 2.0?

---

> ### Author Response · Authors · 2020-11-17
> **Response to AnonReviewer4 (Part 2/2)**
>
> **5. Regarding applying softmax on answer span prediction**
>
> We apply softmax on $s_i$, $e_{i,j}$ before applying the cross-entropy loss. We normalize $e_{i,j}$ across all spans in the passage, to enable the comparison of $(s_i+e_{i,j})$ as scores for all different answer candidates.
>
> **6. Regarding the error in the equation of the MSE loss for predicting K**
>
> You are indeed correct, we use the output of $f_k$ to compute the MSE. We will correct this in the paper.
>
> **7. Regarding the error in the description of $w^o_t$ for question generation**
>
> $w^o_t$ is the probability over the vocabulary with size $D$ at time step $t$. We will correct the statement “output embedding” to “output words probability”.
>
> **8. Further discussion of the UniLM baseline**
>
> We will add further clarification on UniLM in our manuscript.
>
> **9. Regarding generation of unanswerable questions**
>
> We separate the unanswerable and answerable cases in the training set, and then train them separately with independent models as mentioned in *Section 3->Implementation Details*. This is because the mechanism of generating questions may differ in generating answerable and unanswerable questions. For example, the model could exploit a mismatching phrase to make a question plausible but unanswerable. In order to reflect these characteristics, these models were trained separately.
>
> **References**
>
> Yang et al. (2017) : Semi-supervised qa with generative domain-adaptive nets. (ACL 2017)
>
> Golub et al. (2017) : Two-stage synthesis networks for transfer learning in machine comprehension. (EMNLP 2017)
>
> Dong et al. (2019) : Unified language model pre-training for natural language understanding and generation. (NeurIPS 2019)
>
> Alberti et al. (2019) : Synthetic QA corpora generation with roundtrip consistency. (ACL 2019)
>
> Lee et al. (2019) : Latent retrieval for weakly supervised open domain question answering. (ACL 2019)
>
> Lewis et al. (2020) : BART: Denoising Sequence-to-Sequence Pre-training for Natural Language Generation, Translation, and Comprehension (ACL 2020)
>
> Raffel et al. (2020) : Exploring the limits of transfer learning with a unified text-to-text transformer. (JMLR 2020)

---

> ### Author Response · Authors · 2020-11-17
> **Response to AnonReviewer4 (Part 1/2)**
>
> **1. Technical contributions of our paper compared to the related works**
>
> Conventional pre-training approaches tend to be very generic, while our approach is a more appropriate pre-training method focused on the specific task of question generation. To prove the effectiveness of our approach, we compared question generation scores of our pre-training method with Alberti et al. (2019) and Golub et al. (2017) with the same base model architecture, BertGen. Also, we show that our method outperforms Alberti et al. (2019) and Golub et al. (2017) in the downstream MRC scores, even in low-data settings. Furthermore, by applying our method to finetune UniLM (Dong et al. 2019), we show that supplementing the generic pre-training method of UniLM with our focused pre-training is indeed more effective for question generation.
> Lee et al. (2019) and Raffel et al. (2020) utilized Transformers to learn different tasks such as information retrieval and language modeling on large corpora. Yang et al. (2017) introduced a training method for the MRC model by combining synthetic data and human-annotated data. Unlike these methods, our pre-training task for the question generator focuses on generating missing answer-containing sentences in the context to learn representations more suitable to the question generation task. We will add a discussion of these related works.
>
> **2. Regarding comparison of applying ASGen to other models which have a decoder**
>
> We provide a comparison to other pretrained models that have a decoder, such as UniLM (Dong et al., 2019) and BertGen + NS (Alberti et al., 2019).
> BertGen has an encoder-decoder architecture similar to BART (Lewis et al., 2020), and BertGen + NS is pre-trained on the task of generating the next sentence (NS). In *Tables 4 and 5*, BertGen + ASGen demonstrates significantly better question generation scores than BertGen + NS. Our approach also exhibits better downstream MRC scores than using questions from a decoder pre-trained on NS as shown in *Figure 2*.
> Also, we apply our method on UniLM, which has a unified encoder-decoder architecture similar to T5 (Raffel et al., 2020), pre-trained on LM tasks. Our method further improves the question generation capability of this model, as shown in *Tables 4 and 5*.
> It is difficult to provide results of BART and T5 pre-trained with ASGen within the limited rebuttal period due to the large compute requirements for pre-training of these models.
>
> **3. Regarding generalization of our method to other datasets such as Natural Questions**
>
> We have provided the scores of question generation on Natural Questions (short answer) dataset, as shown in *Appendix->Table 10*. Our approach improves the scores in BLEU-4 by $3.8$, METEOR by $2.5$, and ROUGE by $1.1$. We also report improvement on the generation score in MS MARCO (*Table 3*), where questions are collected from user query logs in Bing, similar to Natural Questions. We will clarify this in our main manuscript.
>
> **4. Regarding where the synthetic answers are obtained from during pre-training**
>
> As mentioned in *Section 3->Pre-training Dataset* and in *Appendix E->Details of Wikipedia Processing*, the synthetically generated answers for our pre-training method are obtained by the answer generation model of BertGen, using contexts from Wikipedia. We use the standard pre-trained BERT to initialize the answer generation model, which is further fine-tuned on SQuAD. We will clarify this in the paper.

---

> > ### Author Response · Authors · 2020-11-24
> > **Additional Comment on "2. Regarding comparison of applying ASGen to other models which have a decoder"**
> >
> > **Results of our pre-training method on the T5 model**
> >
> > Pre-training T5 (Small) was relatively quick to train, so we are able to obtain results before the end of the rebuttal period. As shown below, our pre-training step increases question generation scores on SQuAD splits of T5 (Small) using both Small-Wiki and Full-Wiki. We expect our pre-training may show a similar effect in other sized T5 models as well.
> >
> > | Test set on Split 1 | BLEU-4 | METEOR | ROUGE-L |
> > |---------------------------|:------:|:------:|:-------:|
> > | T5 (Small)            |   15.55  |  23.25  |   37.08  |
> > | T5 (Small) + ASGen (Small-Wiki)    |  16.46  |  23.96  |  38.39  |
> > | T5 (Small) + ASGen (Full-Wiki)    |  **16.97**  |  **24.19**  |  **38.85**  |
> >
> > | Test set on Split 2 | BLEU-4 | METEOR | ROUGE-L |
> > |---------------------------|:------:|:------:|:-------:|
> > | T5 (Small)            |   18.77  |  25.22  |   40.46  |
> > | T5 (Small) + ASGen (Small-Wiki)    |  19.18  |  25.88  |  41.29  |
> > | T5 (Small) + ASGen (Full-Wiki)    |  **19.63**  |  **26.05**  |  **41.92**  |
> >
> > The results were obtained using the source code from https://github.com/patil-suraj/question_generation with all default hyper-parameters.

---

> > > ### Comment · AnonReviewer4 · 2020-11-24
> > > **Response**
> > >
> > > Thanks to the authors for the detailed responses.
> > >
> > > It seems I missed a point about UniLM having a pretrained decoder. The added results using T5 small and on BioASQ are also encouraging, hence I am increasing my score to 6. However, there are still weaknesses which prevent me from increasing that score further. First, as reviewer1 points out, the effect on downstream QA performance is only shown for weaker models than the current state of the art. Second, it is still not known whether this pretraining strategy would help over the strongest existing pretrained models (T5-large, BART).

---

> > > > ### Author Response · Authors · 2020-11-25
> > > > **Response to AnonReviewer4**
> > > >
> > > > Thank you for your response to our comments.
> > > >
> > > > As mentioned in Appendix B, we tested our synthetic data with Electra (Large) MRC model which has official code, and its variant is currently in state-of-the-art on single model results for SQuAD (https://rajpurkar.github.io/SQuAD-explorer/). We will further evaluate our pre-training method on the strongest generative models (e.g. other sized T5 or BART).

---

### Official Review · AnonReviewer2 · 2020-11-02

**Rating:** 7
**Confidence:** 4

**Review:**

This paper presents a model for unsupervised pre-training for the task of question generation. The model first predicts the number of answer present in a given paragraph and then selects the top-K answer spans from the paragraph. After selecting the answer spans, the model, then tries to generate the answer containing sentence by inputting the paragraphs less the answer-containing sentence and also the answer span. The key idea is that this unsupervised pre-training strategy is close to the actual task of generating question given the context and the answer. This is also the key differentiator between this work and other existing pre-training strategies for question generation (e.g. Alberti et al 2019).

The paper outperforms other existing methods of question generation on several datasets (two splits of Squad, MS Marco, NewsQA) both on automatic and a small scale human evaluation. Efficacy of the pre-training scheme is shown via the fact that the pre-training scheme also improves other question generation model. Lastly, training a QA model on the synthetic generated questions improves downstream performance of a QA model and the difference is greater in low-data setting suggesting the applicability of this pre-training scheme in low-data regime (although no experiments in a specific low-data domain is reported).

Strengths:
* The pre-training scheme is closer to the original task than other existing methods and can be easily scaled.
* The paper is well written and easy to follow and the experiments and ablations were exhaustive.
* Improvements in low-data QA setting is promising and shows the scope of usefulness of this work

Weaknesses
* I am not sure about the need to predict the number of unique answers in a paragraph or that if it makes sense to do so. I think it is hard to estimate the number of questions that can be generated from a given text and regressing to a value which is present in current datasets might be suboptimal. This is because, the current QA datasets do not ask annotators to generate all possible question from a paragraph. Instead, I believe you could have taken a principled approach of considering named-entities as answers, and trying to generate questions for each entity (or a set of entities together). This would also eliminate the need to predict the number of answer spans in a paragraph at once.
* I think it would be useful for the paper to categorize the kinds of “reasoning” that the generated question requires to answer. For example, is it just fuzzy pattern matching kind of questions or does answering the question require any kind of reasoning (such as multi-hop or numerical reasoning).
* It was not clear to me, how you generate questions which are unanswerable (e.g. those in Squad 2.0)
* Although, it is good to see that training on the synthetically generated questions help in low-data settings, I think the paper would be stronger and more convincing. if an actual experiment was done on a domain which has less data. For example, you could try the BioASQ datasets which have very less annotations and is in the bio-medical domain. It would be interesting to see if pre-training on a scientific corpus is actually helpful. Moreover, some questions in BioASQ need reasoning such as handling list questions, counting and it would be interesting to see if the performance on those questions improve.

Overall, I think the current experiments are reasonably well-done and I think the paper would be much stronger if it was tested on an actual domain which has low-data and also if the paper discusses / categorizes the kind of reasoning that the generated questions require.

---

> ### Author Response · Authors · 2020-11-17
> **Response to AnonReviewer2 (Part 2/2)**
>
> **4. The performance of our method on limited domain data (BioASQ)**
>
> We additionally evaluate our question generation model on BioASQ-factoid data, and test the MRC model on the downstream BioASQ data using synthetic data generated on Wikipedia.
> As shown below, our method improves question generation scores by $6.0$ BLEU-4, $7.8$ METEOR and $6.9$ ROUGE-L. Moreover, even using synthetic data generated on Wikipedia enhances the performance of BERT(Large) by large margin and outperforms BioBERT by $0.95$ Macro F1 (Yes/No) and $1.63$ F1 (List).
> Note that BioBERT is pre-trained on a medical corpus (PubMed) whereas we use a generic corpus. We tested the downstream MRC model on BioASQ by using the official source code from Yoon et al. (2019) and Lee et al. (2019). For the question generation, we test our method on factoid-type 6b. We will update the manuscript as follows:
>
> | Question Generation Model \|| BLEU-4 \|| METEOR \|| ROUGE-L |
> |---------------------------|:------:|:------:|:-------:|
> | BertGen(Large)            |   6.6  |  10.0  |   33.1  |
> | BertGen(Large) + ASGen    |  **12.6**  |  **17.8**  |  **40.0**  |
>
> | BioASQ Models      |\| Factoid (MRR) \|| Yes/No (Macro F1) \||   List-Type (F1)  |
> |--------------------|:-------------:|:-----------------:|:------------:|
> | BERT(Large)        |  34.27(±0.48) |    53.77(±1.72)   | 36.09(±0.55) |
> | BioBERT(Large)   |  **52.34(±0.23)** |    80.10(±1.41)   | 38.12(±0.23) |
> | BERT(Large) + ASGen|  49.17(±0.18) |    **81.05(±1.26)**   | **39.76(±0.29)** |
>
> **References**
>
> Rajpurkar et al. (2016) : Squad: 100,000+ questions for machine comprehension of text. (EMNLP 2016)
>
> Golub et al. (2017) : Two-stage synthesis networks for transfer learning in machine comprehension. (EMNLP 2017)
>
> Subramanian et al. (2018) : Neural models for key phrase detection and question generation. (MRQA at ACL 2018)
>
> Rajpurkar & Jia et al. (2018) : Know What You Don’t Know: Unanswerable Questions for SQuAD. (ACL 2018)
>
> Alberti et al. (2019) : Synthetic QA corpora generation with roundtrip consistency. (ACL 2019)
>
> Yoon et al. (2019) : Pre-trained language model for biomedical question answering. (Joint European Conference on Machine Learning and Knowledge Discovery in Databases 2019)
>
> Lee et al. (2019) : BioBERT: a pre-trained biomedical language representation model for biomedical text mining (Bioinformatics 2019) [-> updated 11/18]

---

> ### Author Response · Authors · 2020-11-17
> **Response to AnonReviewer2 (Part 1/2)**
>
> **1. The reason why we are predicting the number of key answer phrases and using a neural model for extracting answers**
>
> An important thing to do in question generation is figuring out which elements of a given context are interesting. To do this, we predict the number of key answer phrases to obtain a more appropriate set of “answer-like” key phrases that humans tend to choose when they are preparing a question, rather than all possible entity phrases. This tendency can also be found in the SQuAD dataset, which has a varying number of annotated answers per context, even though the annotators were asked to create up to five answers, as shown below. While we do not have golden number of answers for all contexts, this characteristic of SQuAD annotation can still be useful for learning the number of answer candidates.
>
> |   Number of Answers       |   1      |   2      |   3      |   4       |   5       |  6+      |
> |--------------------|----|----|----|-----|-----|----|
> |Percentage of Sample | 0.5 | 0.9 | 9.1 | 21.9 | 60.1 | 7.5 |
>
> The approach of Alberti et al. (2019) & Golub et al. (2017) is limited as they use a fixed-K number of answer candidates or a logit classification on each word, which may not be optimal for selecting answer candidates. As shown in *Table 6*, our dynamic answer prediction module can learn to predict a more appropriate number of "answer-like” key phrases that humans tend to choose than fixed-K or thresholding on answer logit approaches. The results in downstream tasks also demonstrate the validity of our approach even with limited training data (Small-Wiki), as shown in *Appendix->Table 13*.
> Moreover, as mentioned in *Section 5->Related Work->Synthetic Data Generation*, Subramanian et al. (2018) claimed that neural key phrase extraction model is important in identifying interesting candidate answers for the question generation, and that using neural model outperforms entity-tagging baseline and existing rule-based approaches.
>
> **2. Regarding categorization of reasoning for generated questions**
>
> We manually categorized the reasoning type of 150 randomly sampled generated questions on Wikipedia for both answerable and unanswerable types. The results below show that our generated questions often do require multi-hop or other non-trivial reasonings. We used the same categorization as done by SQuAD-v1.1 (Rajpurkar et al., 2016) and SQuAD-v2.0 (Rajpurkar & Jia et al., 2018) and our results are as follows -
>
> | Answerable Question Type            |\|Our Question \|| SQuAD-v1.1 |
> |-------------------------------------|:------------:|:----------:|
> | Lexical Variation (Synonymy)        |     40.7%    |    33.3%   |
> | Lexical Variation (World Knowledge) |     4.0%     |    9.1%    |
> | Syntactic Variation                 |     53.3%    |    64.1%   |
> | Multi Sentence Reasoning            |     21.3%    |    13.6%   |
> | Ambiguous/Unanswerable              |     4.0%     |    6.1%    |
>
>
> | Unanswerable Question Type |\|  Our Question   \| | SQuAD-v2.0 |
> |----------------------------|:------------:|:----------:|
> | Negation                   |     8.0%     |    9.0%    |
> | Antonym                    |     14.7%    |    20.0%   |
> | Entity Swap                |     36.0%    |    21.0%   |
> | Mutual Exclusion           |     9.3%     |    15.0%   |
> | Impossible Condition       |     7.3%     |    4.0%    |
> | Other Neutral              |     19.3%    |    24.0%   |
> | Answerable                 |     5.3%     |    7.0%    |
>
> Note that each example can be assigned to multiple types for answerable questions. We will update our manuscript with these changes.
>
> **3. Regarding how we generate unanswerable questions**
>
> We separate the unanswerable and answerable cases in the training set, and then train them separately with independent models as mentioned in *Section 3->Implementation Details*. This is because the mechanism of generating questions may differ in generating answerable and unanswerable questions. For example, the model could exploit a mismatching phrase to make a question plausible but unanswerable. In order to reflect these characteristics, these models were trained separately.

---

> > ### Comment · AnonReviewer2 · 2020-11-23
> > **Response**
> >
> > Thank you for doing the experiment on BioASQ and doing the categorization of questions. These are super helpful and they make the paper a lot stronger. I am recommending acceptance.
> >
> > Regarding predicting the number of answers: I agree that predicting answer phrases is important. My point is there could be a better way for doing that than formulating it as a regression task. This is because of the same reason as you point out: annotators were asked to generate a fixed number of questions. Instead, generating questions about entities or key concepts present in the text seems more principled to me.

---

> > > ### Author Response · Authors · 2020-11-24
> > > **Response to AnonReviewer2**
> > >
> > > Thank you for your response to our comments.
> > >
> > > The reason why we formulated this as a regression task is that we needed a mechanism to select a subset of generated answer candidates, and we use weak supervision of the number of answers to this end. As an alternative, we also experimented with “Fixed-K” and "Thresholding on answer logit" but their performance was lower than “Dynamic-K”. Note that all the entities from the context are often present in the list of answer candidates from which all these approaches select.

---

### Author Response · Authors · 2020-11-19
**Update of submission based on the reviewer's feedback**

We have uploaded a revised version of the paper, addressing the reviewers' concerns. We thank the reviewers for their feedback. Below is a summary of changes to the paper -

* Categorization of reasoning type of generated questions is added in *Section 4. Experimental Results->4.4. Qualitative Analysis of Question Generation->Categorization of Reasoning Type* and *Tables 10, 11*.
* Added reference to limited-data domain BioASQ in *Section 3. Experimental Setup->Benchmark Datasets* and detailed results of transfer learning on BioASQ in *Appendix F* and *Table 17*.
* Downstream MRC results for comparison of pre-training method added in *Section 4. Experimental Results->4.2 Downstream MRC task performance* and *Table 8*.
* Added measures of statistical significance for Dynamic-K  in downstream results in *Appendix D* and *Table 16*.
* Added discussion on predicting the number of answers in *Section 2.1 Dynamic Answer Prediction* and *Appendix K*. [updated 11/24]
* Added discussion regarding Electra scores in *Appendix B*.
* Added clarification on technical contribution and discussion about missed references in *Section 5. Related Works*.
* Added detailed procedure of generating unanswerable question in *Appendix J*. [updated 11/24]
* Added details of UniLM settings in *Section 3. Experimental Setup -> Implementation Details*.
* Clarified procedure of obtaining synthetic answers in *Section 3. Experimental Setup->Pre-training Dataset*.
* Corrected $f_k$ for MSE loss and description of output probabilities $w^o_t$ in equations in *Section 2*.

---

> ### Author Response · Authors · 2020-11-24
> **Additional update of submission with T5 results**
>
> We have uploaded a revised version of the paper, addressing the reviewers' concerns regarding application of ASGen to T5.
>
> * Added T5, T5 + ASGen question generation scores to *Table 2* and *Appendix G* and *Table 18*.

---

### Decision · Program_Chairs · 2021-01-07
**Final Decision**

**Decision:**

Reject

**Comment:**

All reviewers appreciate the good quality of this submission with a good idea and solid execution (as said by R3). The paper is clearly written and the addition during the discussion have greatly improved it as acknowledged by all reviewers.

However, a major weakness of the submission still needs to be addressed before a publication at ICLR.

As said in the paper, the task of question generation is a task whose main impact is to improve downstream tasks, and primarily QA. The evaluations follow that and extra-experiments (e.g. BioASQ) and discussion wrt state of the art (e.g. Alberti et al.) reinforce them. Yet, as pointed out by R1 & R4, the effect on downstream QA performance is only shown for weaker models than the current state of the art (e.g. T5, BART). Since the rebuttal period was not long enough to run these experiments, it is impossible to assess how the proposed approach compare to them with the current draft. Adding the experiments on T5 (Small) is a step in the right direction but it is not enough for that. Without those experiments, one can not conclude that this pretraining strategy would also help over the strongest existing pretrained model.

The authors should run those experiments to make the arguments presented in the submission much stronger.